# Cognitive Functioning in Patients with Pediatric-Onset Multiple Sclerosis, an Updated Review and Future Focus

**DOI:** 10.3390/children6020021

**Published:** 2019-02-04

**Authors:** Joy B. Parrish, Emily Fields

**Affiliations:** 1Department of Neurology, Jacobs School of Medicine and Biomedical Sciences at the University at Buffalo, 1001 Main St., Buffalo, NY 14203, USA; 2Department of Psychology, Roosevelt University, 430 S Michigan Ave, Chicago, IL 60605, USA; efields01@mail.roosevelt.edu

**Keywords:** multiple sclerosis, demyelinating disorder, pediatric, cognitive, neuropsychology, psychosocial

## Abstract

Pediatric-onset multiple sclerosis (POMS) is relatively rare, but as technology and neuroimaging advance, an increasing number of cases are identified, and our understanding of how multiple sclerosis (MS) impacts the developing brain improves. There are consistent findings in the literature highlighting the impact of MS and other demyelinating diseases on cognitive functioning and cognitive development. We also have a better understanding of how POMS impacts psychosocial functioning and functional outcomes in daily living. This paper hopes to review findings associated with cognitive and psychosocial functioning in patients with POMS, as well as explore more recent advances in the field and how they relate to cognitive and psychosocial outcomes. We also discuss the ongoing need for future studies with a focus on better understanding deficits and disease correlates, but also preventative measures and potential rehabilitation.

## 1. Introduction

Multiple sclerosis (MS) is an inflammatory neurodegenerative disease primarily affecting young adults. Approximately 3–5% of all individuals with MS experience disease onset before the age of 18 (POMS) [1,2,3]. The past decade has provided advancement in understanding disease characteristics, diagnosis, treatment effectiveness, and longitudinal outcomes. Studies have also revealed a better understanding of cognitive and psychosocial outcomes in patients with POMS.

MS is a chronic inflammatory disease of the central nervous system (CNS), with progressive neurodegeneration. The mechanism of neurodegeneration is still not fully understood, but there is evidence to suggest an interplay of oxidative stress, mitochondrial injury, and ion channel dysfunction secondary to inflammation causing neurodegeneration, while modifying factors such as age, sex, and genetic factors contribute to severity and course [4].

The classic definition for clinical definite MS requires clinical and/or laboratory support for an ongoing process with proof of dissemination in time and space. The International Pediatric MS Study Group has published operational definitions for POMS [5,6], which are based on MS diagnostic criteria in adults (McDonald Criteria) [7,8]. The most recent consensus definition paper for POMS [9] continues to support diagnosis based on neuroimaging findings, but lesion findings including dissemination in time and space radiologically may not be enough, and added emphasis is put on clinically relevant history. There has been increased focus on additional biomarkers for distinct demyelinating disease phenotypes. Anti-myelin oligodendroglial glycoprotein (MOG) antibodies have been identified in up to one third of children with a first acute episode of inflammatory demyelination and are predictive of non-MS diseased (e.g., ADEM). However, persistent anti-MOG antibodies are associated with MS, although less than one quarter of POMS patients evidence this biomarker [10,11]. The presence/absence of the anti-MOG antibodies, depending on time during the disease course, can help clinicians to better understand treatment needs and prognosis.

POMS primarily presents as a relapsing remitting type (over 90% of cases), and there tends to be a higher T2 lesion burden, supporting the idea that POMS has a more inflammatory pathobiology [12]. Younger patients with MS are also more likely to have seizures and brainstem and cerebellar involvement [13]. Although relapses may be more frequent in POMS, recovery tends to be better, and there is a slower rate of progression of disease, potentially related to greater plasticity in the developing nervous tissue [14,15]. A longitudinal study from Italy [16] examined 97 patients over an average of 12 years and showed that the majority of patients had Expanded Disability Status Scale (EDSS) scores that remained below 4 (89%), but 40% had worsening disability. Despite the slower rate of progression, given the early age of onset, disability often occurs earlier in adulthood. As with adults, there may be a link between MS and reduced vitamin D levels, Epstein–Barr virus infection, genetics, obesity, as well as smoking (or second-hand exposure) [17,18,19]. Several recent studies have also shown evidence of pre- and post-natal exposure to environmental toxins (e.g., household chemicals (pesticides)), associated with increased risk for POMS [20].

There have been many studies examining cognitive functioning in patients with POMS, with the majority of studies finding that approximately one third of patients suffer from cognitive impairment. The following review serves to further identify cognitive deficits seen in patients with POMS, how cognitive functioning is assessed in this population, and disease and environmental correlates of cognitive functioning outcomes. Literature was reviewed from sources including PubMed search, research cited in related POMS articles, and previously acquired articles.

## 2. Cognitive and Psychosocial Outcomes

### 2.1. Cognitive Functioning Outcomes in POMS

Assessment of cognitive functioning in POMS continues to advance, with ongoing improvement in assessment batteries, understanding of disease correlates, and additional focus on risk factors and protective features. Research has consistently found that approximately one third of pediatric patients with MS suffer from neurocognitive impairment. This is somewhat less than what is typically reported in the adult literature, which ranges from 50% [21,22,23,24] to close to 70% [25,26]. Early diagnosis of cognitive dysfunction can be extremely important in POMS, given that physical status can be well preserved. Children also appear to be more vulnerable to cognitive impairment early on in the disease process.

Areas of neurocognitive impairment in POMS often mimic those in adults, including processing speed, working memory, and visual spatial processing; however, there have been additional findings of impact on language, which is not typical in the adult cohort [27,28,29,30,31,32]. Poor verbal knowledge has implications in outcomes of education and general functional attainment. It may be associated with direct impact of disease process on learning and language development, as well as a secondary interference from high absenteeism from school due to medical complications and impact of other neurocognitive defects impacting learning (attention, processing speed, etc.). Nevertheless, language impairment is not always reported in patients with POMS. One recent study examining a large number of patients from several clinics in the United States reported language as the least frequent domain showing impairment in their cohort [24].

Early longitudinal studies found evidence of cognitive decline in patients with POMS [29,33,34]. An initial Italian study was startling, with findings suggesting worsening of cognitive functioning in 70% of their POMS patients over only 2 years, despite relatively stable physical and disease status [34]. Further evaluation, extending these results to 5 years in 48 patients, showed there was some evidence of improvement in 67% of patients from year 2 to year 5 [35]. Comparisons between baseline and 5-year follow-up assessments showed 56% of patients declined on their impairment index, 25% improved, and close to 19% remained stable. Declines were associated with male sex, younger age of onset, and lower education. The US Pediatric MS Network findings suggest a limited decline in cognitive functioning over a 1–2-year time span [36]. Another recent study examining patients and controls over a 1-year period found that controls generally showed greater improvement than patients, and 25% (7/28) of patients showed clinically significant decline [37]. They suggest that the lower rate of improvement may be indicative of lack of age-appropriate development. Ongoing longitudinal analysis will help to clarify the severity and frequency of cognitive decline, as well as provide insight regarding effective intervention strategies and disease-related correlates that may help in preventing or reducing the disease impact on cognitive functioning.

### 2.2. Psychosocial and Functional Outcomes

We assume, and to some degree know, that neurocognitive deficits and physical disability lead to functional impairment (e.g., poor academic progress and need for school accommodations, inability to participate in sports). The adult literature continues to show us that cognitive decline and disability level, together with progressive course, influence social and work-related function in adults with early-onset MS [38,39]. However, scientific evaluation of functional outcomes in children is limited. Functional outcomes tend to be harder to identify and quantify in children. In adults, vocational placement or employment is often used. In children, we know school is the primary functional outcome to examine, but there can be many reasons why a child fails in school other than cognitive impairment. There are some initial findings showing the effects of POMS on school functioning, such as high absenteeism, need for special education services, and being held back or dropping out of school [27,28,32]. Our center has consistently shown that approximately two thirds of patients (approximately *n* = 66) receive or have received a combination of therapeutic intervention (occupational, physical, or speech/language therapies) and/or require special education plans or have academic accommodations in place at their schools (personal communication). The majority require classroom accommodations (e.g., extended time on exams, reduced workload, providing extra support during longer absences), with younger patients typically requiring more therapeutic intervention (occupational, physical, speech/language therapy).

We also know that there are several additional factors to cognitive impairment, such as fatigue, depression, and poor quality of life, which can impact functional outcomes, regardless of MS progression. Measurement of mood and fatigue and understanding their impact on cognitive and daily functioning in patients with MS has been more consistent in the adult literature [40,41]. Recent focus has increased in the pediatric population, but further examination is necessary.

Quality of life (QoL) assessment can help to determine the impact of cognitive impairment and disease burden on daily functioning. Literature examining the assessment of QoL in POMS is limited, with only a few studies to date discussing the topic. Their findings suggest significant reductions in health-related QoL (HRQOL) scores despite short disease duration and generally fair physical ability [42]. Others have examined QoL as a tertiary measure and have found evidence of poorer quality of life [43] but provide only a limited assessment of the correlation between QoL and cognitive impairment. One study examining young adults with pediatric-onset MS found, not surprisingly, that physical health-related QoL was related to EDSS, while depression was related to the Mental index of health related quality of life (HRQOL), but overall, there was not a significant reduction in HRQOL compared to controls [44]. Fatigue and depression have also been shown to impact QoL in POMS [45]. 

Studies exploring rates of fatigue in children with MS reported it as occurring in 20–75% of patients [43,46,47,48,49]. A small number of studies have examined the association between fatigue and cognitive functioning. One smaller study (*n* = 26) found that POMS is associated with fatigue and emotional difficulties, which were related to executive dysfunction [50]. Goretti and colleagues [33] found that fatigue was associated with elevated self-reported depression symptoms. Rater differences were evident, with self-reported cognitive fatigue associated with impaired problem solving, while parent-reported cognitive fatigue was associated with impaired verbal learning, cognitive flexibility, and comprehension. By contrast, several authors have found minimal evidence for a relationship between subjective fatigue (either self- or parent-reported) and objective cognitive functioning [51,52,53].

Depression is another common comorbidity in patients with MS that has been minimally evaluated in pediatric-onset MS. Fatigue and depressive symptoms often tend to overlap. In our group, we found that a quarter of pediatric patients with demyelinating disorder (e.g., acute disseminated encephalomyelitis (ADEM), MS) had elevated parent-reported symptoms of depression and self-reported fatigue, and there was a higher rate of fatigue than depression in child self-report [48]. Other studies have shown similar findings [43,54]. MacAllister et al. [28] found that depression was present in half of the cases, while Goretti and colleagues found that 17% of patients based on self-report and 30% based on clinical interview were classified as having an affective disorder [55]. An Italian group found more drastic rates of significant fatigue reported, with nearly 75% of children with MS reporting fatigue, while only a small percentage (6%) reported depression. Within this sample, over half reported that MS had negative effects on their everyday life and school [32]. Another study examining comorbid psychiatric diagnoses and cognitive functioning in POMS found that those with a psychiatric diagnosis had a higher rate of cognitive impairment [56]. Interestingly, cognitive functioning was found to predict the presence of clinical problems (e.g., anxiety, somatization) on self- and parent-reported behavioral assessments (behavior assessment system for children—second edition (BASC-2); [51]).

Increased focus on evaluation and treatment of fatigue and depression in children with demyelinating disorders is essential. We have learned from the adult literature how prevalent comorbid psychiatric disorders and fatigue are in patients with MS, and how symptoms associated to fatigue and psychiatric conditions can significantly impact disease and functional outcomes [57,58,59]. There is extensive literature reviewing the detrimental effects of depression on academic, social, and vocational functioning. We continue to need to increase evaluation of fatigue and symptoms of psychiatric disorder (depression, anxiety, personality change) in addition to physical outcomes and symptoms, even in young children. Moreover, analysis of the effectiveness of interventions such as psychotherapy methods (e.g., cognitive behavioral intervention), behavioral activation, group supportive counseling, educational groups, and pharmacological intervention is direly needed in this population.

Social functioning in children with POMS has also been an interesting area of recent research. With a general population increase in the rate of social functioning disorders in children (e.g., social communication disorder, autism spectrum disorder), it is interesting to look at this area of functioning in patients with POMS and how disease may impact social skill development. Charvet and colleagues (2014) examined social cognition, defined as the cognitive processes governing social situations, such as theory of mind, in a small group of patients with POMS. They found that patients with POMS performed worse on measures of theory of mind compared to controls, suggesting poorer ability to process social information [60]. Findings were associated with cognitive dysfunction and disease characteristics (number of relapses and disease duration). Additional evaluation of the direct impact of disease on social functioning, as well as the indirect effect (missing time from school/social events, cognitive impairment, academic difficulties, inability to participate in certain group activities, etc.) will be interesting.

## 3. Assessment of Cognitive Functioning in POMS

The lack of a universally accepted standard neuropsychology test battery for POMS has limited the ability to compare and combine data across centers. Despite differences in batteries, consistent evidence of approximately 30% of POMS patients showing cognitive impairments suggests that findings are not just test-specific [32,36,46]. Measures used in batteries consist primarily of already established neuropsychological measures with sound psychometric properties. Specific details regarding psychometric properties for individual tests can be found in measurement manuals. Several centers in the US have collaborated to develop a core battery of tests (The National Multiple Sclerosis Society Consensus Neuropsychological Battery for Pediatric MS: NBPMS) [61]. There is an additional establishment of regression-based norms for this battery, which help to improve sensitivity [62,63]. However, the battery is limited by its lack of measures within specific domains (e.g., problem solving) and the length of time needed to complete testing. Additional evaluation of validity and reliability of screening measures or brief batteries is ongoing. One study found good results in detecting impairment in POMS using only four measures (the Brief Neuropsychological Battery for Children (BNBC)), which examines expressive vocabulary (WISC, vocabulary), cognitive flexibility (trail making test—part B), processing speed (SDMT), and learning and memory (selective–reminding test) [64]. This battery showed high sensitivity (96%) and specificity (76%) in detecting cognitive impairment in patients with POMS. It was suggested that this may be a good screening battery. The International Pediatric MS Study Group has also suggested a shorter core and a supplemental full neuropsychological battery [65]. There was preliminary evidence for the efficacy of using the symbol digits modality test (SDMT) as a screening measure to assess for cognitive dysfunction [66], as it has been found effective in the adult MS population [67]. However, findings have not been replicated. Interestingly, another study using a similar task but computer based (c-SDMT) found that difference in total time to complete the task did not differ between POMS patients and controls, but that POMS patients were less likely to show faster performance across all successive eight trials compared to controls. This suggests reduced capacity for procedural learning [68] and may be an aspect of the test that could be used to screen for cognitive impairment. 

Additional examination of brief batteries includes a recent study, which examined the efficacy of using a 15-min battery (Brief International Cognitive Assessment of MS: BICAMS) including measures of visual learning (brief visuospatial memory test—revised: BVMT-R), verbal learning (Rey auditory verbal learning test: RAVLT), and attention (SDMT), compared to a 15-min computer-administered battery of speeded processing tasks (Cogstate brief battery) and found similar detection rates of cognitive dysfunction across batteries [69]. Similarly, Bartlett et al. (2018) compared Cogstate and BICAMS outcomes in patients with POMS and found that the Cogstate composite score revealed slowing in the POMS group compared to controls, while the BICAMS composite did not differentiate the groups [70]. Interestingly, other disease-related measures may also help in identifying/screening for cognitive impairment in patients with POMS. Brief batteries and computer-based assessments may be more time efficient but also have drawbacks. This is especially true in children, where an assessment of intellectual reasoning is important to determine general cognitive development (e.g., IQ), as this can impact performance across all domains and skew findings. However, IQ assessments (e.g., Wechsler scales) are often lengthy. Most adult batteries use a premorbid assessment of IQ, such as word reading, but this is a poor predictor of general functioning, especially in young children, and reading skills are still developing. Parent education has been used to estimate intellectual reasoning, but this too can be confounded by several factors, including socioeconomic status (SES) and cultural differences. 

Further evaluation of computer-based assessment and/or other brief batteries with regard to reliability and validity is needed, but initial findings are promising in offering greater access to assessment and potentially early intervention, which may improve functional outcomes.

## 4. Disease Characteristics Associated with Cognitive and Psychosocial Outcomes

### 4.1. MS Treatment and Cognitive Functioning

It is believed that if there is a reduction in MS-related relapses, there should be associated lower lesion load, decreased brain atrophy, and thus, less cognitive impairment. Studies examining the impact of treatment efficacy on cognitive outcomes are limited in POMS. The focus has been on treatment safety and disease outcomes, but little focus has been given to cognitive and psychosocial outcomes, as they relate to treatment, and/or any direct effects of treatment on cognitive functioning. Most disease-modifying therapies (DMTs) have not gone through complete testing in randomized, placebo-controlled clinical trials for patients with pediatric-onset MS and are being prescribed “off-label” by clinicians. As these studies are progressing, it is important to address if approaches of treating pediatric MS patients have any long-term impact on patients, in particular, on the physical, cognitive, developmental, and social outcomes of the children. The typical first-line treatments for POMS include interferon-beta (IFN-B) and glatiramer acetate (GA) [71]. Research examining effects of intramuscular interferon beta-1a have shown improvement or stability on neuropsychological testing post-treatment in adults with MS [72,73,74,75]. One study showed little cognitive decline in adult-onset MS patients treated with glatiramer acetate [76], and several showed improvement in cognitive functioning on natalizumab [77]. Research examining how therapeutic interventions directly impact cognitive functioning in POMS is difficult to conduct and equally difficult to interpret due to small sample sizes, widely varying treatment methods, and variable disease severity and course. In a large Italian study, with 331 patients, treatment with interferon beta-1a did not impact quality of life, depression or fatigue over a three-year period [78].

### 4.2. Neuroimaging

Neuroimaging findings and their relationship with neurocognitive status in patients with MS have been part of a constantly changing landscape over the past several years. Initially, it was believed that lesion load was most important in predicting neurocognitive outcomes. Then, grey matter (GM) atrophy showed greater correlations with cognitive outcomes, and even more specifically, findings of thalamic atrophy were associated with neurocognitive status. Volume reduction at the onset in POMS with further reduction over time is similar to adult findings and concerning when considering neurocognitive outcomes [31,79,80,81,82,83,84,85,86,87,88,89,90].

The thalamus, known as a “relay station”, has been linked to several areas of cognitive functioning given its bidirectional connections with several brain regions. Mesaros et al. [79] identified thalamic tissue loss in POMS consistent with adult findings [80,81], but more general cortical gray matter loss was not found. Additional studies have shown evidence of broader cortical volume loss. One study examining POMS patients and healthy controls found that patients had a smaller head size and decreased normalized brain volume. Moderate correlations were found between T2 lesion load and disease duration with thalamic and brain volume [82]. After controlling for these factors, they also found that thalamic volumes were lower than in controls, raising the important inquiry as whether the phenomenon is secondary to tissue loss/atrophy versus delayed/impeded growth. Recently, Weier and colleagues (2015) [83] identified failure of cerebellar volume to reach age-expected trajectories, as well as cerebellar white matter volume decreases in pediatric MS patients compared to healthy controls over a three-year follow up study. A longitudinal analysis of regional brain volumes found failure of age-normative brain growth in the MS group, especially in the thalamus. T2 lesion volume was correlated with greater reduction in age-expected thalamic volume [84].

Analyses examining the association between neuroimaging and cognitive functioning in patients with POMS have shown several interesting findings. One study found cognitive impairment in approximately 30% of patients, with thalamic and corpus callosum volume being associated with cognitive status [31]. Thalamic volume was the most robust neuroimaging finding, along with global and region volume, correlated with cognitive functioning, while lesion volume was not as strongly correlated. These findings are similar to data in the adult-onset MS population linking cognitive impairment to neurodegeneration and impact on gray matter structures. Nevertheless, findings suggesting white matter integrity and its importance in cognitive ability have also been demonstrated [85,86]. Others have found relevance of other subcortical structure integrity. Rocca and colleagues (2016) found that hippocampal volume, often associated with learning and memory processes, was reduced in patients with POMS compared to controls. Regional hippocampal volume, not overall hippocampal volume, was associated with T2 lesion volume, attention, and language abilities. Global hippocampal volume did not differentiate between intact and cognitive impaired patients; however, particular regions of the hippocampus (subiculum, dentate gyrus subfields of right hippocampus) were different between these groups [87]. Another group [88] found that amygdala volume was smaller in patients compared to controls and poorer memory was associated with reduced amygdala volume. Amygdala volume was also associated with parent-reported adaptive skills (social skills, communication). More common posterior involvement seen in children with POMS has also been shown to be associated with cognitive impairment. Rocca and colleagues (2014) [89] found cognitively impaired patients (*n* = 16; 45%) had higher occurrence of T2 lesions and more severe damage to white matter and grey matter (as measured by atrophy and diffusivity abnormalities) in the posterior regions of parietal lobes.

Evaluation of diffusion tensor imaging (DTI) in patients with POMS has provided additional insights into correlates between structural integrity and functioning. Several studies have found differences in DT-MRI between patients with POMS and healthy controls [90,91,92]. An interesting study examining new onset POMS patients with imaging analysis at first demyelinating event found that patients had reduced fractional anisotropy (FA) and increased diffusion of the bilateral superior longitudinal fasciculus and corpus callosum, areas often linked with processing speed and integrating information/processes, compared to healthy controls. There was an additional finding that expected developmental changes over time were not seen in patients [90]. There has been limited examination of DTI and cognitive functioning, but Bartlett et al. (2019), using Cogstate and BICAMS, found that the Cogstate composite showed correlations with regional FA and correlated significantly with uncinated fasciculus FA. BICAMS only showed weak to moderate correlations with FA [70]. De Meo and colleagues [92] found general abnormalities in patients with POMS on DT-MRI, as well as differences in increased activation of the left thalamus, anterior insula, and anterior corpus callosum (ACC) and decreased recruitment of the right precuneus on fMRI compared to healthy controls during a continuous performance test (assessing sustained attention). Those with cognitive impairment showed decreased recruitment in several regions and more severe structural damage of white matter tracts connecting those regions.

Functional neuroimaging findings are limited in the pediatric population. An early study [93] found patients with POMS showed supplementary recruitment of the left sensorimotor cortex during simple movements along with decreased activation in functional connectivity between this area and the left thalamus, insula, secondary sensorimotor cortex, and supplemental motor area. Larger activation areas responsible for specific activities, compared to those in healthy individuals, including deep gray structures (particularly the thalamus), were postulated to explain increased fatigue and possibly early damage seen in the thalamus as “an overworked hub” [94]. A more recent study by Rocca and colleagues (2014) [95] showed that patients with POMS had decreased functional connectivity in several regions compared to controls, as well as increased functional connectivity in the right medial frontal gyrus, which was correlated with T2 lesion volume. Moreover, POMS patients with cognitive impairment had additional abnormalities in functional connectivity. Examination of functional neuroimaging and information processing speed in POMS patients, as measured by the SDMT, showed a correlation between faster response time and greater activation of several areas (right inferior occipital, anterior cingulate, right superior parietal, thalamus, and left superior occipital cortices) and decreased overall activation of the right middle frontal gyrus during the task, even with no difference in performance on the measure (response time or accuracy) [96]. Faster response time was also associated with greater activation in the left superior occipital region in the MS group only, suggestive of attenuated activation in frontal regions in POMS, even with similar performance outcomes.

Resting state fMRI studies have also yielded evidence of differences between patients with POMS and controls and are a promising focus of future functional examination given increased access and ease of use. Akbar and colleagues (2016) found higher resting state functional connectivity (FC) in POMS patients in several regions (precuneus, anterior cingulate cortex, frontal medial cortex, and cerebellum). Greater T2 lesion volume and lower thalamic volume were associated with a reduced FC of the thalamus. Further correlational analysis with cognitive functioning showed the FC of the left frontal medial cortex was negatively correlated with the cognitive composite. They suggest that there is greater recruitment during resting state in patients with reduced cognitive efficiency [97].

As neuroimaging advances, we will continue to gain a better understanding of structural and functional differences in patients with POMS, as well as their implication/predictive value in disease and cognitive outcomes. Additional focus on neurodevelopment will be crucial, as POMS occurs during a period of ongoing brain development. Understanding how the disease impacts this development and related functional outcomes, identifying neuroprotective mechanisms, and longitudinal assessment will be imperative.

## 5. Other Risk Factors of Cognitive Impairment

Several studies have begun to look at risk factors and/or contributing factors to outcomes in POMS. Review of cognitive impairment in adult onset MS (AOMS)outlined several risk factors for cognitive impairment, including psychiatric distress, pain, headache, and fatigue [98,99]. These risk factors are not unique to MS and can impact neurocognitive functioning regardless of medical condition.

Obesity has been found to be associated with increased risk of acquiring POMS, particularly during adolescence and in females [100,101,102,103,104]. Unfortunately, obesity has risen substantially over the past several years. As further evidence of this complex relationship, Gianfrancesco and colleagues (2017) found that low vitamin D concentrations in serum and increased BMI independently contributed to increased likelihood of pediatric-onset MS after controlling for covariates (e.g., sex, genetic ancestry). Another study found that obesity in girls was associated with significantly increased risk of MS/clinically isolated syndrome (CIS), as well as increased risk of transverse myelitis [101]. The mechanism of and relationship between obesity and neurologic disease, including MS, have been discussed in an article by Lee and Mattson [105].

Recent literature provides supporting evidence of impaired cognitive functioning in obese or overweight children, as well as changes in brain structure and function in this population [106]. Often, studies show an association between poor executive function and obesity. A review [107] found several studies linking childhood obesity with poor performance on inhibitory tasks, as well as a link between early poor inhibitory control and later higher body mass index (BMI). A recent meta-analysis found that obesity was associated with worse overall functioning, including areas of attention, cognitive flexibility, working memory, reward response and delayed gratification [108]. A Turkish study found that not only was cognitive functioning lower in children with obesity, but they also had higher reported anxiety symptoms compared to controls [109].

Neuroimaging examination in children with obesity helps in understanding functional differences that occur that may contribute to findings. One study by Maayan and colleagues found that obese adolescents perform worse on inhibitory control tasks, which was associated with smaller orbitofrontal cortex volume [110]. Functional changes were also noted in the dorsolateral prefrontal cortex prior to and following meals in obese children [111,106]. It remains unclear if cognitive or structural/functional changes cause or contribute to obesity (e.g., less control leads to overeating) or if they are a result of obesity [112]. Moreover, obesity is a risk factor for obstructive sleep apnea (OSA), which has been associated with cognitive dysfunction [113], which can also be contributory in cognitive outcomes. OSA and related hypoxic events can increase oxidative stress, induce chronic inflammation, and cause changes in the blood-brain barrier (BBB), which can directly impact brain integrity and function [114] and be additive to the inflammation seen in MS patients.

There are no studies to date that have specifically examined obesity/BMI and cognitive outcomes in patients with POMS. Given the increased risk for MS in adolescents who are overweight, as well as the findings of increased cognitive and psychosocial dysfunction in children with obesity, this should be an area of focus for future studies. This will allow us to better understand the interaction between obesity and cognitive outcomes in patients with POMS, but also, and potentially more importantly, help in addressing possible preventative measures to reduce not only occurrence or relapses in MS but also progression of cognitive deficit in patients with POMS (e.g., diet and physical activity regimes).

There are several additional risk factors for more adverse cognitive outcomes in patients with POMS. The African-American race was found to be associated to more severe disease course, as well as more negative neurocognitive outcomes in POMS [115,116]. Lower socioeconomic status (SES) is also associated with poorer neurocognitive functioning [117,118]. SES can impact health directly due to poor accessibility to health care, limited knowledge about and adherence to treatment, and variable nutrition. Genetic risk factors and interactions between several risk factors (second-hand smoke, genetic predisposition, physical functioning) were also evaluated [119,120,121]. A recent pilot study [26] searched for genetic signatures associated with cognitive impairment in patients with POMS. They found that expressions of 11 miRNAs associated with cognitive test performance. Some genes found have previously been identified as associated with cognitive dysfunction (e.g., BST1, NTNG2, SPTB, STAB1), while the expression of several miRNAs was correlated with regional brain volumes, particularly the cerebellum. Biomarkers, such as genetic findings, may help to predict those at highest risk for cognitive impairment, as well as to help to better understand the mechanisms that contribute to cognitive dysfunction in patients with POMS. Further examination of these and other environmental factors and how they contribute to disease course and functional outcomes is needed.

## 6. Cognitive Rehabilitation and Preventative Measures

As mounting evidence shows, a substantial number of patients with POMS have cognitive deficits and/or psychosocial difficulties. Therefore, added focus needs to be placed on preventative measures and rehabilitation to either improve or help to compensate for areas of deficit. Protective factors, including cognitive reserve, personality, and response shift have been evaluated in adults [61,122,123]. There is one study that suggests higher cognitive reserve, as estimated by IQ, may also be a protective factor against cognitive impairment in POMS [124]. Understanding how to assess or potentially even improve cognitive reserve in children is still in its infancy, but an intriguing concept worthy of further investigation.

Rehabilitative therapies for cognition typically include those oriented towards improvement in memory, language, attention, and executive function [125]. A study in AOMS suggests mild improvement in sustained attention with computer intervention but limited self-reported differences in daily functioning [126]. Other studies in AOMS have found benefits with cognitive training and rehabilitation techniques, including increased brain activation and improved performance on a working memory task (paced auditory serial addition test: PASAT) [54,126], as well as improved learning and memory [127] and attention [128]. Given the effectiveness of computerized training programs and behavioral intervention techniques demonstrated in the adult population, focus on early intervention in POMS is even more justified. There have been only a few small pilot studies evaluating cognitive rehabilitation in POMS [129,130]. There is preliminary evidence that specific computer-assisted cognitive rehabilitation of attention may improve global cognitive functioning in patients with POMS [129]. In a very small study (*n* = 5), Hubacher and colleagues found lasting improvement in working memory in 2/5 patients (out to 9 months), with associated initial increases in network activation and inter-network connectivity (fMRI) [130]. There are preliminary data in other patient populations that cognitive rehabilitation interventions are effective, particularly with improving attention [131,132]. If programming can use technologies of interest (video games, electronics), adherence would likely be greater, and there would be fewer demands on clinicians. As with any intervention strategy, adherence and generalizability of effects is of concern, and further evaluation with larger cohorts is needed.

Interventions to address comorbidities in POMS, like obesity, require medical, psychological, and psychosocial interventions, as well as general changes to public health. Chitnis and colleagues mentioned success with group programs for adolescents with MS (e.g., teen adventure program) [3]. Further consideration of pharmacological treatment for fatigue, attention problems, and mood (methylphenidate, antidepressants) is important but has yet to be evaluated in patients with POMS. Studies in adults have shown some improvement in cognitive functioning with pharmacological intervention [133,134]. Physical and occupational therapy (PT, OT) interventions to address motor rehabilitation as well as visual spatial processing and motor integration are needed. Sandroff and colleagues reviewed how exercise/physical activity affects cognition in patients with MS and found evidence to support a positive effect [135]. Physical functioning and gait variability (variation in mean step time), but not gait speed, were also found be associated with cognitive outcome in patients with POMS with minimal disability (EDSS mean of 1.6) [136] and may be able to be used as an early clinical marker of cognitive performance.

The most common way children receive therapeutic and other such services is through school interventions and accommodations as provided in 504 Plans or Individualized Educational Programs (IEPs). Assistance from neuropsychologists can be crucial in helping children to receive needed interventions. Psychotherapeutic interventions, including simple behavioral activation as well as more specific cognitive–behavioral techniques, can improve mood, social functioning, and self-evaluation. Research to determine the effectiveness of specific techniques in patients with POMS will help to guide treatment recommendations, but implementation of already established, symptom-specific interventions should be used. A recent review of behavioral interventions in adult-onset MS suggests improvement in fatigue, depression, and motor functioning, as well as improvement in disease progression (reduced relapse rate, new lesions) following group education or face-to-face behavioral intervention [137], which may be related to stress reduction. Unfortunately, there has been no systematic evaluation of the effects of established therapeutic interventions in patients with POMS.

## 7. Conclusions and Future Perspective

Future studies should continue to focus on assessing and understanding specific areas of cognitive deficit in patients with POMS and the longitudinal impact on development. Added focus on developing reliable screening mechanisms or understanding disease correlates and the impact on cognitive functioning will be important in increasing access to care. However, potentially even more importantly, focus should also be redirected to understanding risk factors (e.g., cognitive reserve, SES, environmental factors, obesity, personality), studying intervention strategies (e.g., pharmacological, cognitive–behavioral, and psychoeducational interventions), and determining interventions to reduce risk associated with increased disease severity and progression of cognitive decline (e.g., diet, physical activities, cognitive rehabilitation, counselling). We also need to consider techniques and interventions that incorporate interests of children and adolescents to improve compliance (e.g., social media, technology for scheduling and reminding, and video games and interactive technologies (Wii) for exercise and cognitive rehabilitation). Additional assessment and understanding of how more general factors, such as SES, parent education, personality, and family history, play a role in outcomes need to be considered during clinical evaluation and research examination. Progress in patient care depends not only on evaluating disease outcomes and DMTs, but also on how to treat the “whole” patient, including cognitive, behavioral, and psychosocial wellbeing.

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
