# Peer review of "Cognitive Functioning in Patients with Pediatric-Onset Multiple Sclerosis, an Updated Review and Future Focus"

_children, 2019, doi:10.3390/children6020021_

Reviewer 1 Report

This paper provides an updated review of cognitive functioning in patients with pediatric MS. It is extremely well written and organized, and gives a comprehensive review of the field and future recommendations. Only a few minor edits are suggested:

Page 2, line 46: authors list adult MS risk factors that are also found in POMS. Obesity should also be included here, given that it comes up further into the manuscript.

Page 2, Section 3: The manuscript would flow much better if section 3 was placed right after the introduction. Section 2 seems to awkwardly fit between section 1 and 3. If possible, I'd recommend reordering and moving the "assessment of cog. functioning" elsewhere.

Page 7, line 327: typo-- "growing areA of focus."

Author Response

Response: Reviewer 1

Thank you for taking time to review this article and provide valuable feedback.  Below is a detailed response to your comments/suggestions:

Page 2, line 46: authors list adult MS risk factors that are also found in POMS. Obesity should also be included here, given that it comes up further into the manuscript.

Obesity was added to this sentence.

Page 2, Section 3: The manuscript would flow much better if section 3 was placed right after the introduction. Section 2 seems to awkwardly fit between section 1 and 3. If possible, I'd recommend reordering and moving the "assessment of cog. functioning" elsewhere.

Thank you for this comment.  The section was moved as suggested.

Page 7, line 327: typo-- "growing areA of focus."

The typo was corrected. 

Thank you again for your comments and suggestions. 

Sincerely,

Joy Parrish, Ph.D., ABPP

Board Certified in Clinical Neuropsychology

Board Certified Subspecialist in Pediatric Neuropsychology

Clinical Associate Professor

Department of Neurology

Jacobs School of Medicine and Biomedical Sciences at the University at Buffalo

Reviewer 2 Report

It is an interesting manuscript that addresses the cognitive impairment in patients with pediatric onset multiple sclerosis (POMS). The authors review the available studies that performed neuropsychological evaluations of POMS patients. In relation to cognitive dysfunction, the authors report the available cognitive batteries, as well as the confounding factors (i.e., fatigue and affective symptoms), the risk factors (e.g., obesity),  the available and potential interventions, and the neural underpinning (i.e., neuroimaging data). I have some comments/suggestions.

Abstract: The ‘s’ may be omitted in advances.

Introduction:

-  It would be helpful to introduce the cognitive deficits in POMS since the following paragraphs will be about this aspect of the disease.

-  The introduction could benefit from adding a sentence reminding the readers about the pathophysiological mechanisms of MS.

- The introduction could benefit from adding the relationship between POMS and environmental, maternal and perinatal exposures. The following references could be helpful:

o   Mar S, Liang S, Waltz M, Casper TC, Goyal M, Greenberg B, Weinstock-Guttman B, Rodriguez M, Aaen G, Belman A, Barcellos LF, Rose J, Gorman M, Benson L, Candee M, Chitnis T, Harris Y, Kahn I, Roalsted S, Hart J, Lotze T, Moodley M, Ness J, Rensel M, Rubin J, Schreiner T, Tillema JM, Waldman A, Krupp L, Graves JS, Waubant E; U.S. Network of Pediatric Multiple Sclerosis Centers. Several household chemical exposures are associated with pediatric-onset multiple sclerosis. Ann Clin Transl Neurol. 2018 Oct 9;5(12):1513-1521. doi: 10.1002/acn3.663. eCollection 2018

o   Graves JS, Chitnis T, Weinstock-Guttman B, Rubin J, Zelikovitch AS, Nourbakhsh B, Simmons T, Waltz M, Casper TC, Waubant E; Network of Pediatric Multiple Sclerosis Centers. Maternal and Perinatal Exposures Are Associated With Risk for Pediatric-Onset Multiple Sclerosis. Pediatrics. 2017 Apr;139(4). pii: e20162838. doi: 10.1542/peds.2016-2838.

o   Pétrin J, Fiander M, Doss PMIA, Yeh EA. A Scoping Review of Modifiable Risk Factors in Pediatric Onset Multiple Sclerosis: Building for the Future. Children (Basel). 2018 Oct 26;5(11). pii: E146. doi: 10.3390/children5110146.

-  It is also of interest to remind the reader of the available treatments:

o   Ghezzi A, Amato MP, Makhani N, Shreiner T, Gärtner J, Tenembaum S. Pediatric multiple sclerosis: Conventional first-line treatment and general management. Neurology. 2016 Aug 30;87(9 Suppl 2):S97-S102. doi: 10.1212/WNL.0000000000002823.

-  Please move the period after the reference “involvement.[10]”

-  It would be helpful to add information on the selection criteria and the consulted databases for this review.

Section 2:

-  Please add a reference for l. 49-52.

-  BICAMS is a worldwide used battery (Brief International Cognitive Assessment for MS) to evaluate cognitive functions in MS, and the abbreviation merits to be defined upon its first appearance. The same applies to the name of the tests used to assess visual learning (BVMT-R) and verbal  learning (RAVLT).

-  The authors may want to add more details on the Brief Neuropsychological Battery for Children (reference [19]).

-  The authors are encouraged to add more details on studies that employed Weschler Intelligence Scale or its abbreviated versions when assessing intelligence quotient (IQ) in POMS patients.

-   It would be also of interest to mention the psychometric properties of the considered scales/batteries.

-  The manuscript does not discuss social cognition which is an important cognitive facet that seems to be altered in patients with POMS:

o   Charvet L, Cleary R, Vazquez K, Belman A, Krupp L. Social cognition in pediatric-onset multiple sclerosis (MS). Multiple Sclerosis Journal, 2014; 20(11):1478–1484. doi:10.1177/1352458514526942

Section 3:

-   Recent works (such as the following ones) reviewed the adult MS literature and reports cognitive dysfunction in up to 65-70% of patients:

o   Langdon DW. Cognition in multiple sclerosis. Curr Opin Neurol. 2011 Jun;24(3):244-9. doi: 10.1097/WCO.0b013e328346a43b.

o   Chalah MA, Ayache SS. Deficits in Social Cognition: An Unveiled Signature of Multiple Sclerosis. J Int Neuropsychol Soc. 2017 Mar;23(3):266-286. doi: 10.1017/S1355617716001156.

-   Please define the abbreviation ADEM (Acute disseminated encephalomyelitis) in l. 156.

-   Please define the abbreviation BASC-2 (Behavior Assessment System for Children-Second Edition) in l. 168

-   In l. 187, please define DMTs (disease modifying therapies)

-   In l. 191 please define IM IFN b (intramuscular interferon) and abbreviate instead in l. 197

-   Please remove the ‘ in DMT’s : “Research examining how DMT’s impact cognitive functioning in POMS is difficult to conduct”

Section 4:

-   Please provide a reference for l. 200-206

-   In l. 239, please define the abbreviations GM (grey matter) and WM (white matter)

-  When exposing the involved neural regions (e.g., thalamus, corpus callosum, posterior parietal regions, precuneus, anterior cingulate cortex, frontal medial cortex, cerebellum, and the WM fasciculi), it would be interesting to add some sentences linking the involvement of these structures in cognitive functions.

Section 5:

-   Please define the abbreviation CIS (clinically isolated syndrome) in l. 300

-   Please substitute “are” for “area” in l. 327

-   When discussing the risk factors, it is of interest to elaborate more on genetic risk factors and their relationship with cognitive deficits. The authors may want to review the following paper on this matter:

o   Liguori M, Nuzziello N, Simone M, Amoroso N, Viterbo RG, Tangaro S, Consiglio A, Giordano P, Bellotti R, Trojano M. Association between miRNAs expression and cognitive performances of Pediatric Multiple Sclerosis patients: A pilot study. Brain Behav. 2019 Jan 17:e01199. doi: 10.1002/brb3.1199.

Section 6:

-   In l. 355, please define the PASAT (Paced auditory serial addition test)

-   In l. 377 the authors may want to substitute “effects” for “affects”

Author Response

Response: Reviewer 2

Thank you for taking time to review this article and provide valuable feedback.  I appreciate your thorough review.  Below is a detailed response to your comments/suggestions:

Abstract: The ‘s’ may be omitted in advances

The  ‘s’ was omitted

Introduction:

-    It would be helpful to introduce the cognitive deficits in POMS since the following paragraphs will be about this aspect of the disease.

-   The introduction could benefit from adding a sentence reminding the readers about the pathophysiological mechanisms of MS.

 The introduction could benefit from adding the relationship between POMS and environmental, maternal and perinatal exposures. The following references could be helpful:

o   Mar S, Liang S, Waltz M, Casper TC, Goyal M, Greenberg B, Weinstock-Guttman B, Rodriguez M, Aaen G, Belman A, Barcellos LF, Rose J, Gorman M, Benson L, Candee M, Chitnis T, Harris Y, Kahn I, Roalsted S, Hart J, Lotze T, Moodley M, Ness J, Rensel M, Rubin J, Schreiner T, Tillema JM, Waldman A, Krupp L, Graves JS, Waubant E; U.S. Network of Pediatric Multiple Sclerosis Centers. Several household chemical exposures are associated with pediatric-onset multiple sclerosis. Ann Clin Transl Neurol. 2018 Oct 9;5(12):1513-1521. doi: 10.1002/acn3.663. eCollection 2018

o   Graves JS, Chitnis T, Weinstock-Guttman B, Rubin J, Zelikovitch AS, Nourbakhsh B, Simmons T, Waltz M, Casper TC, Waubant E; Network of Pediatric Multiple Sclerosis Centers. Maternal and Perinatal Exposures Are Associated With Risk for Pediatric-Onset Multiple Sclerosis. Pediatrics. 2017 Apr;139(4). pii: e20162838. doi: 10.1542/peds.2016-2838.

o   Pétrin J, Fiander M, Doss PMIA, Yeh EA. A Scoping Review of Modifiable Risk Factors in Pediatric Onset Multiple Sclerosis: Building for the Future. Children (Basel). 2018 Oct 26;5(11). pii: E146. doi: 10.3390/children5110146.

 It is also of interest to remind the reader of the available treatments:

o   Ghezzi A, Amato MP, Makhani N, Shreiner T, Gärtner J, Tenembaum S. Pediatric multiple sclerosis: Conventional first-line treatment and general management. Neurology. 2016 Aug 30;87(9 Suppl 2):S97-S102. doi: 10.1212/WNL.0000000000002823.

-   Please move the period after the reference “involvement.[10]”

-   It would be helpful to add information on the selection criteria and the consulted databases for this review.

Introduction: additional information regarding cognitive deficits in patients with POMS, as well as the general pathophysiological mechanisms of MS were added.  Additional references and associated information regarding environmental exposures were added as suggested.  Treatment information was put into the section about treatment, as it seems this was more prudent, but can be moved or repeated if deemed necessary.  The period was moved to after the citation.  A short sentence was added regarding how the information for the review article was obtained. 

Section 2. Please add a reference for l. 49-52.

-        BICAMS is a worldwide used battery (Brief International Cognitive Assessment for MS) to evaluate cognitive functions in MS, and the abbreviation merits to be defined upon its first appearance. The same applies to the name of the tests used to assess visual learning (BVMT-R) and verbal  learning (RAVLT).

-  The authors may want to add more details on the Brief Neuropsychological Battery for Children (reference [19]).

 The authors are encouraged to add more details on studies that employed Weschler Intelligence Scale or its abbreviated versions when assessing intelligence quotient (IQ) in POMS patients.

-   It would be also of interest to mention the psychometric properties of the considered scales/batteries.

  The manuscript does not discuss social cognition which is an important cognitive facet that seems to be altered in patients with POMS:

o   Charvet L, Cleary R, Vazquez K, Belman A, Krupp L. Social cognition in pediatric-onset multiple sclerosis (MS). Multiple Sclerosis Journal, 2014; 20(11):1478–1484. doi:10.1177/1352458514526942

Section 2 (now section 3): References were added as requested.  Abbreviations used in the text were defined upon first appearance.  Greater detailed information was added regarding BNBC, assessing intellectual reasoning (Wechsler scales) and an short section on social cognition was also added.  I did not include the specific psychometric properties of all measures mentioned, as I felt this would greater deter from the flow of the paper, especially for those not familiar with reliability and validity coefficients.  I instead added a comment about the general reliability of the measures used and directed readers to either identified reference of measure itself for specific on psychometric properties.  All measures identified in this review have sound psychometric properties. 

Section 3 (now section 2): Recent works (such as the following ones) reviewed the adult MS literature and reports cognitive dysfunction in up to 65-70% of patients:

o   Langdon DW. Cognition in multiple sclerosis. Curr Opin Neurol. 2011 Jun;24(3):244-9. doi: 10.1097/WCO.0b013e328346a43b.

o   Chalah MA, Ayache SS. Deficits in Social Cognition: An Unveiled Signature of Multiple Sclerosis. J Int Neuropsychol Soc. 2017 Mar;23(3):266-286. doi: 10.1017/S1355617716001156.

-   Please define the abbreviation ADEM (Acute disseminated encephalomyelitis) in l. 156.

-   Please define the abbreviation BASC-2 (Behavior Assessment System for Children-Second Edition) in l. 168

-   In l. 187, please define DMTs (disease modifying therapies)

-   In l. 191 please define IM IFN b (intramuscular interferon) and abbreviate instead in l. 197

-  Please remove the ‘ in DMT’s : “Research examining how DMT’s impact cognitive functioning in POMS is difficult to conduct”

Suggested references and information regarding cognitive impairment in adults was added.  Acronyms were defined, as listed, and the typo of DMT’s was changed to DMTs. 

Section 4:

Please provide a reference for l. 200-206

 In l. 239, please define the abbreviations GM (grey matter) and WM (white matter)

  When exposing the involved neural regions (e.g., thalamus, corpus callosum, posterior parietal regions, precuneus, anterior cingulate cortex, frontal medial cortex, cerebellum, and the WM fasciculi), it would be interesting to add some sentences linking the involvement of these structures in cognitive functions.

References were added as requested.  Acronyms were again defined.  Information regarding regional associated with specific cognitive functioning was added as suggested. 

6. Section 5: Please define the abbreviation CIS (clinically isolated syndrome) in l. 300

-   Please substitute “are” for “area” in l. 327

-   When discussing the risk factors, it is of interest to elaborate more on genetic risk factors and their relationship with cognitive deficits. The authors may want to review the following paper on this matter:

o   Liguori M, Nuzziello N, Simone M, Amoroso N, Viterbo RG, Tangaro S, Consiglio A, Giordano P, Bellotti R, Trojano M. Association between miRNAs expression and cognitive performances of Pediatric Multiple Sclerosis patients: A pilot study. Brain Behav. 2019 Jan 17:e01199. doi: 10.1002/brb3.1199.

a. The acronym CIS was defined and the typo “are” for “area” was changed.  The reference and associated information regarding genetic findings associated with cognitive outcomes was added as suggested. 

7. Section 6: In l. 355, please define the PASAT (Paced auditory serial addition test)

-  In l. 377 the authors may want to substitute “effects” for “affects” 

 The acronym for PASAT was defined.  The typo (effects/affects) was corrected.

Thank you again for your comments and suggestions, and your time taken to review this article. 

Sincerely,

Joy Parrish, Ph.D., ABPP

Board Certified in Clinical Neuropsychology

Board Certified Subspecialist in Pediatric Neuropsychology

Clinical Associate Professor

Department of Neurology

Jacobs School of Medicine and Biomedical Sciences at the University at Buffalo

Reviewer 3 Report

p.p1 {margin: 0.0px 0.0px 0.0px 0.0px; font: 12.0px 'Helvetica Neue'; color: #0433ff; -webkit-text-stroke: #0433ff} p.p2 {margin: 0.0px 0.0px 0.0px 0.0px; font: 12.0px 'Helvetica Neue'; color: #0433ff; -webkit-text-stroke: #0433ff; min-height: 14.0px} p.p3 {margin: 0.0px 0.0px 0.0px 0.0px; font: 12.0px Helvetica; color: #0433ff; -webkit-text-stroke: #0433ff} p.p4 {margin: 0.0px 0.0px 0.0px 0.0px; font: 12.0px Helvetica; color: #000000; -webkit-text-stroke: #000000; min-height: 14.0px} p.p5 {margin: 0.0px 0.0px 0.0px 0.0px; font: 12.0px Helvetica; color: #000000; -webkit-text-stroke: #000000} span.s1 {font-kerning: none} span.s2 {font-kerning: none; color: #000000; -webkit-text-stroke: 0px #000000} span.s3 {font-kerning: none; color: #0433ff; -webkit-text-stroke: 0px #0433ff}

The paper by Parrish & Fields is generally very informative, well written and updated. I have some suggestions as well as some minor, typographic and orthographic, concerns:

Introduction

The readers of “Children” would appreciate having some additional informations on potential biomarkers of the disease which can reflect pathogenic mechanisms and therapeutic strategies. A brief mention to MOG-based distinct clinical phenotypes related to the age of onset and patterns of disease course would be useful.

3.2. Psychosocial and Functional Outcomes

line 126-129 Our center has consistently shown... (omissis). I would encourage the authors to be a bit more detailed, e.g. how many paediatric MS patients require special education plans or academic accommodation in your center? which one do they need the most?

4.1. MS Treatment and Cognitive Functioning

Also, it would be worth some few lines on the effect of treatment (e.g interferon-beta) on the quality of life, fatigue and depression in paediatric MS.

Stylistic make-up

line 182 It is believed that if there is a reduction in MS related relapses (MS-related)

line 327 should be a growing are of focus (not clear sentence).

line 333 African-American race, was (please delete comma)

line 334 Lower socioeconomic status (SES), is (please delete comma)

Author Response

Response: Reviewer 3

Thank you for taking time to review this article and provide valuable feedback.  Below is a detailed response to your comments/suggestions:

Introduction

The readers of “Children” would appreciate having some additional informations on potential biomarkers of the disease which can reflect pathogenic mechanisms and therapeutic strategies. A brief mention to MOG-based distinct clinical phenotypes related to the age of onset and patterns of disease course would be useful.

Introduction:  Additional general information regarding biomarkers (e.g., anti-MOG antibodies, etc) and associated disease course was added.

3.2. Psychosocial and Functional Outcomes

line 126-129 Our center has consistently shown... (omissis). I would encourage the authors to be a bit more detailed, e.g. how many paediatric MS patients require special education plans or academic accommodation in your center? which one do they need the most?

Section 3 (now section 2): More information regarding our specific site and services provided to our patients.

4.1. MS Treatment and Cognitive Functioning

Also, it would be worth some few lines on the effect of treatment (e.g interferon-beta) on the quality of life, fatigue and depression in paediatric MS.

Section 4: A comment was already in the text regarding how certain treatments have been found to be associated with cognitive and psychosocial outcomes (lines 254-257).  Unfortunately there is very limited information available, as this has not often been described in outcome studies.  If there is additional information you are aware of, I would be happy to add it to this manuscript. 

Stylistic make-up

line 182 It is believed that if there is a reduction in MS related relapses (MS-related)

line 327 should be a growing are of focus (not clear sentence).

line 333 African-American race, was (please delete comma)

line 334 Lower socioeconomic status (SES), is (please delete comma

General style, etc: all deletions and additions were made as suggested.  An added comment was inserted to clarify the sentence in original line 327. 

Thank you again for your comments and suggestions, and your time taken to review this article. 

Sincerely,

Joy Parrish, Ph.D., ABPP

Board Certified in Clinical Neuropsychology

Board Certified Subspecialist in Pediatric Neuropsychology

Clinical Associate Professor

Department of Neurology

Jacobs School of Medicine and Biomedical Sciences at the University at Buffalo